# Doping Ferrocene-Based Conjugated Microporous Polymers with 7,7,8,8-Tetracyanoquinodimethane for Efficient Photocatalytic CO_2_ Reduction

**DOI:** 10.3390/molecules29081738

**Published:** 2024-04-11

**Authors:** Shenglin Wang, Qianqian Yan, Hui Hu, Xiaofang Su, Huanjun Xu, Jianyi Wang, Yanan Gao

**Affiliations:** 1Key Laboratory of Ministry of Education for Advanced Materials in Tropical Island Resources, Hainan University, No 58, Renmin Avenue, Haikou 570228, China; wangshenglin@hainanu.edu.cn (S.W.); yanqianqian@hainanu.edu.cn (Q.Y.); sxf@hainanu.edu.cn (X.S.); energywang@hainanu.edu.cn (J.W.); 2School of Science, Qiongtai Normal University, Haikou 571127, China; xuhuanjun86@iccas.ac.cn

**Keywords:** conjugated microporous polymer, ferrocene, TCNQ, doping, photocatalytic CO_2_ reduction

## Abstract

The design and synthesis of organic photocatalysts remain a great challenge due to their strict structural constraints. However, this could be mitigated by achieving structural flexibility by constructing permanent porosity into the materials. Conjugated microporous polymers (CMPs) are an emerging class of porous materials with an amorphous, three-dimensional network structure, which makes it possible to integrate the elaborate functional groups to enhance photocatalytic performance. Here, we report the synthesis of a novel CMP, named TAPFc-TFPPy-CMP, constructed by 1,1′3,3′-tetra(4-aminophenyl)ferrocene (TAPFc) and 1,3,6,8-tetrakis(4-formylphenyl)pyrene (TFPPy) monomers. The integration of the *p*-type dopant 7,7,8,8-tetracyanoquinodimethane (TCNQ) into the TAPFc-TFPPy-CMP improved the light adsorption performance, leading to a decrease in the optical bandgap from 2.00 to 1.43 eV. The doped CMP (TCNQ@TAPFc-TFPPy-CMP) exhibited promising catalytic activity in photocatalytic CO_2_ reduction under visible light, yielding 546.8 μmol g^−1^ h^−1^ of CO with a selectivity of 96% and 5.2 μmol g^−1^ h^−1^ of CH_4_. This represented an 80% increase in the CO yield compared to the maternal TAPFc-TFPPy-CMP. The steady-state photoluminescence (PL) and fluorescence lifetime (FL) measurements reveal faster carrier separation and transport after the doping. This study provides guidance for the development of organic photocatalysts for the utilization of renewable energy.

## 1. Introduction

Covalent organic polymers (COPs) are porous framework materials composed of organic monomers containing non-metallic, lightweight elements such as C, N, O, B, and Si that have been cross-linked by covalent bonds [1,2]. These materials can be categorized based on their crystallinity and pore structures into various types, including covalent triazine frameworks (CTFs) [3,4], conjugated microporous polymers (CMPs) [5,6], hyper-cross-linked polymers (HCPs) [7,8], polymers of intrinsic microporosity (PIMs) [9,10], covalent organic frameworks (COFs) [11,12], porous aromatic frameworks (PAFs) [13,14], porous organic cages [15,16], among others. Specifically, CMPs can be produced using a variety of organic building blocks with straightforward synthetic processes. CMPs feature easily altering energy levels without compromising porosity, a high specific surface area, and excellent thermostability and chemical stability [17,18,19]. In terms of molecular structure and/or function, the rigid structure of building blocks and the integration of conjugated units into CMPs provide structural support for microporous channels, distinguishing them from conjugated small molecules or linear conjugated polymers [20,21]. This feature allows CMPs to retain some photovoltaic properties of conjugated polymers while preserving porosity, which allows for accommodating. Furthermore, CMPs have shown essential applications in gas adsorption and separation, catalysis, semiconductor materials, energy storage, and chemical sensing due to their ability to easily achieve chemical modification and functional regulation.

The narrow bandgap of conjugated polymers (usually 1.5–2.5 eV) allows electrons to be excited from the highest occupied molecular orbital (HOMO) to the lowest occupied molecular orbital (LUMO), which facilitates charge injection, leading to the creation of semiconducting materials [22]. In their undoped state, conjugated polymers lack free charges, resulting in low electrical conductivity. To enhance their conductivity, doping is necessary to introduce charge carriers by transferring electrons or holes to and from the conjugated backbone. Doping with appropriate *n*-type or *p*-type additives is a well-established method to adjust the energy levels of molecular orbitals in functional materials, thereby regulating charge carrier injection [23,24,25]. Molecular dopants with low ionization energy (IE) act as electron donors to facilitate *n*-doping by providing electrons to the LUMO level of the maternal acceptor. Conversely, dopants with high electron affinity (EA) function as electron acceptors and induce *p*-doping by extracting electrons from the HOMO level of the maternal donor [26]. *p*-type doping of conjugated polymers has been explored to develop conductive and magnetic organic polymers [27,28,29], since the pioneering research on conductive polyacetylene films [30]. Open-shell polymers can be formed through charge transfer complexation when electron-rich polymers are doped with *p*-type dopants like tetracyanoethylene, 7,7,8,8-Tetracyanoquinodimethane (TCNQ), and their derivatives [31,32]. It remains a great challenge to obtain *n*-type doping in organic semiconductors due to the typically shallow LUMO levels. Electron donors with a powerful capability to donate electrons to organic semiconductors are scarce, and these donors are prone to being unstable in the air, limiting their use in industrial applications.

Ferrocene (dicyclopentadienyliron, Fc) is often referred to as the ‘benzene’ of modern organometallic chemistry due to its significance in the field. First synthesized in 1951, Fc marked a milestone as the first pure hydrocarbon derivative of iron [33,34,35]. Since then, a variety of derivatives of Fc have been designed and synthesized, known for their exceptional stability and unique chemical properties. These derivatives have garnered much interest in different disciplines, such as asymmetric catalysis, nonlinear optics, and electrochemistry, owing to the quasi-reversible oxidation of iron(II) [36,37,38]. Fc derivatives are recognized for their excellent electron-donating properties, diverse structural types, broad light absorption range, and fast charge transfer rate. Consequently, Fc-based functional materials are widely utilized as efficient and stable electron donors in various optoelectronic applications [39]. The construction of CMPs with Fc derivatives as monomers can enrich the complexity of CMPs and show good promise in organic semiconductors, particularly when doped with *p*-type TCNQ to produce unique charge transfer complexes. While most reported Fc-based CMPs feature a disubstituted Fc group as the connecting unit, CMPs with a highly symmetrical tetrasubstituted Fc group are less common [40,41,42,43].

In this study, we synthesized 1,1′3,3′-tetra(4-aminophenyl)ferrocene (TAPFc) monomer and reacted it with 1,3,6,8-tetrakis(4-formylphenyl)pyrene (TFPPy) to create an electron-rich CMP (named TAPFc-TFPPy-CMP) through the Schiff-base condensation reaction (Figure 1). By doping with TCNQ, the optical bandgap of the TAPFc-TFPPy-CMP was obviously decreased. The mechanism study showed that faster carrier separation and electrotransport were achieved after the doping of TCNQ. As a result, the doping significantly enhanced the photocatalytic performance of TAPFc-TFPPy-CMP. The TCNQ@TAPFc-TFPPy-CMP yielded 546.8 μmol g^−1^ h^−1^ of CO with a selectivity of 96% and 5.2 μmol g^−1^ h^−1^ of CH_4_, demonstrating an 80% increase in the CO yield compared to the undoped TAPFc-TFPPy-CMP.

## 2. Results and Discussion

### 2.1. Synthesis and Characterization

TAPFc-TFPPy-CMP was synthesized through the Schiff base condensation reaction between the tetrasubstituted TAPFc and four-armed TFPPy under solvothermal conditions. The tetrahedral structure of TAPFc and the high-density linkages afforded TAPFc-TFPPy-CMP rich porosity, giving enough room to accommodate guest molecules. To account for the doping of TCNQ into the CMP, a model compound was also synthesized (see Supporting Information). The formation of TAPFc-TFPPy-CMP was confirmed by Fourier transform infrared (FT-IR) spectra (Figure 1a). Specifically, the peak corresponding to the aldehyde (C=O) bond at 1691 cm^−1^ and the peaks owing to the amino (N-H) bond at 3310 cm^−1^ and 3209 cm^−1^ decreased significantly, while a new peak attributed to the imine bond (C=N) concurrently emerged at 1620 cm^−1^, resembling that of the model compound. These changes in FT-IR spectra indicated the successful condensation reaction between TAPFc and TFPPy.

Furthermore, field emission scanning electron microscopy (FE-SEM) revealed the rod-shaped morphology of TAPFc-TFPPy-CMP (Figure 1b). The porosity of the material was assessed by N_2_ adsorption isotherm analysis at 77 K (Figure 1c). The isothermal curves represent Type I adsorption isotherms as classified by the International Union of Pure and Applied Chemistry (IUPAC). When the relative pressure P/P_0_ > 0.4, the desorption isotherm in the middle does not coincide with the absorption isotherm, resulting in a hysteresis loop. This indicates that the lamellar structure creates pores, or the presence of mesopores in the material, leading to capillary condensation. The material has a Brunauer–Emmett–Teller (BET) surface area of 1085 m^2^ g^−1^ and a pore volume of 0.60 cm^3^ g^−1^. The predicted pore size distribution curve based on the Barrett–Joyner–Halenda (BJH) model is shown in Figure 1d, TAPFc-TFPPy-CMP, with an average pore diameter of 2.1 nm.

### 2.2. Doping with TCNQ

The doping with TCNQ was first carried out in the model compound with 1:2 and 1:4 molecular ratios, respectively. As expected, the FT-IR spectra displayed a distinct -CN peak around 2176 cm^−1^ (Appendix A), while the UV/VIS diffuse reflectance spectroscopy (UV–Vis DRS) exhibited a widened absorption range (Appendix A). The optical bandgap was thus reduced from 2.05 to 1.57 and further to approximate 1.5 eV when increasing the molecular ratio from 1:2 to 1:4 (Appendix A), indicating that the Fc structural units could interact with TCNQ to form a stable charge transfer complex. Following the successful doping of the model compound with TCNQ, we proceeded to investigate the doping of the CMP with TCNQ. The doping conditions were first screened, including solvent, temperature, and the molecular ratio of TCNQ. Similarly, the -CN peak of TCNQ-doped TAPFc-TFPPy-CMP (named TCNQ@TAPFc-TFPPy-CMP) at 2174–2177 cm^−1^ in FT-IR spectra was all observed in THF, MeCN, DCM, DMF, and DOX (Appendix A). It is evident that different UV–Vis DRS spectra were found in these solvents (Appendix A). Among these solvents, TCNQ@TAPFc-TFPPy-CMP in MeCN exhibited a broader absorbance and the corresponding lowest optical band gap of 1.42 eV. Thus, MeCN was chosen as the optimal solvent for the following exploration. The effect of temperature, durations, and molecular ratio on the UV–Vis DRS and optical bandgap was systematically investigated (Appendix A). Considering the boiling point of MeCN and the chemical stability of TAPFc-TFPPy-CMP, the optimal doping condition was determined as follows: temperature of 60 °C, duration of 2 days, and TCNQ dosage of 1:4.

Doping leads to a color deepening of the CMP from red to dark brown under optimal conditions (insert in Figure 2a). The FT-IR and UV–Vis DRS results are consistent with the model compound. The absorption spectrum of TCNQ@TAPFc-TFPPy-CMP extends below 900 nm, encompassing the UV–visible and NIR regions, whereas TAPFc-TFPPy-CMP absorbs strongly below 700 nm (Figure 2a). Doping results in a broader absorption spectrum, indicating a significant π–π interaction between TCNQ and the material, along with the appearance of a charge transfer band between 600 and 700 nm, which accounts for the color deepening post-doping. The presence of a broad, low-intensity absorption in the near-infrared region around 1250 nm is a characteristic feature of conjugated and doped materials, signifying the distribution of free-radical charge carriers and the formation of mixed valence bands between doped and undoped regions [44]. The Tauc curve analysis reveals a considerable reduction in the bandgap of the doped material from 2.00 to 1.43 eV, facilitating electron transitions and leading to a higher intrinsic carrier concentration (Figure 2b). The shift in the absorption band edge is crucial for enhancing the material’s absorption across the spectrum. Doped materials can effectively utilize the entire visible spectral range compared to undoped materials. Additionally, a narrower band gap necessitates less excitation energy, which is advantageous for photogenerated carrier excitation, transitions, and separation. The presence of -CN peaks at approximately 2174 cm^−1^ in the FT-IR spectrum (Figure 2c) confirms the success of the doping process. Additionally, the peak corresponding to -CN at 2221 cm^−1^ of TCNQ shows a noticeable redshift. This shift is mainly due to the decrease in electron cloud density resulting from the π–π interaction between TAPFc-TFPPy-CMP and TCNQ, causing an expansion of the conjugated system. There was no significant displacement or weakening observed in the imine bond peaks, providing further evidence of the stability of the CMP structure. This was also evident in the XPS spectra (Figure 2d), where a new -CN peak at 398.5 eV appeared after TCNQ doping, while the imine bond peaks remained consistent before and after the doping. The BET-specific surface area of the doped material decreased from 1085 to 787 m^2^ g^−1^ (Appendix A). The decrease in specific surface area suggests that the material has undergone doping with TCNQ. The N_2_ adsorption isotherm at 77 K shows a consistent type I adsorption isotherm, indicating that the pore structure of the material has not been altered. Although a high dosage of TCNQ (1:4) was used, only a slight decrease in pore size (from 2.1 to 1.9 nm) was observed (Appendix A), indicating the efficient doping of TCNQ into the CMP interlayer without obviously sacrificing the pores. Additionally, thermogravimetric analysis (TGA) showed that TCNQ@TAPFc-TFPPy-CMP has a composition temperature of about 419 °C, which is much higher than that of TCNQ (260 °C), suggesting that TCNQ was strongly chemically adsorbed within the CMP instead of simply physically mixing (Appendix A). The minor decrease in thermal stability following TCNQ doping is caused by the thermal disintegration of TCNQ units existing in the material.

### 2.3. Photocatalytic Performance

The impact of doping on material properties was initially investigated through a photocatalytic CO_2_ reduction reaction. Mott–Schottky measurements revealed that the TAPFc-TFPPy-CMP and TCNQ@TAPFc-TFPPy-CMP exhibited flat-band potentials of −0.75 and −0.86 V vs. Ag/AgCl, respectively (see Appendix A). Furthermore, the positive slope of the Mott–Schottky plots indicated that both materials are *n*-type semiconductors. It is known that for *n*-type semiconductors, the LUMO energy level is equal to the flat band potential [45], resulting in LUMO energy levels of −0.55 and −0.66 eV (vs. NHE) for TAPFc-TFPPy-CMP and TCNQ@TAPFc-TFPPy-CMP, respectively. According to the band gap, the corresponding HOMO energy levels were calculated to be 1.45 and 0.77 eV (vs. NHE). Energy band diagrams were depicted for both materials, revealing that their LUMO energy level potentials are lower than those of the CO_2_/CO redox potential (Appendix A). This suggests that the photocatalytic conversion of CO_2_ to CO is thermodynamically favorable.

The photocatalytic CO_2_ reduction reaction was carried out under visible light (300 W Xe lamp, λ ≥ 400 nm). Typically, 10 mg of catalyst was sonicated for 20 min and suspended in a solvent mixture of acetonitrile, H_2_O, and triethanolamine (3:1:1). The reaction system was then purified with pure CO_2_ to produce CO and CH_4_ as the primary products. The yields of CO and CH_4_ were 303.4 μmol g^−1^ h^−1^ and 5.2 μmol g^−1^ h^−1^, respectively, with a 94% selectivity for CO after 4 h of reaction catalyzed by TAPFc-TFPPy-CMP. On the other hand, the TCNQ@TAPFc-TFPPy-CMP catalyst led to a CO yield of 546.8 μmol g^−1^ h^−1^ with 96% CO selectivity and CH_4_ yield of 6.1 μmol g^−1^ h^−1^ (Figure 3a). Obviously, an 80% increase in the CO yield was observed after the doping of the CMP with TCNQ, suggesting that doping significantly enhanced the catalytic performance of the Fc-based CMPs. Moreover, both materials exhibited excellent cycling stability, as no obvious changes in the FT-IR spectrum were observed. Their catalytic activities remained almost unchanged after five cycles (Figure 3b).

To investigate the impact of dopants on the photocatalytic performance of materials, we examined carrier separation and transfer efficiency through the measurements of steady-state photoluminescence (PL) and fluorescence lifetime (FL). The PL spectra indicate a significant decrease in the fluorescence intensity of the CMP after doping TCNQ (Figure 4a), suggesting suppression of photogenerated electron–hole pair complexation and enhanced carrier separation efficiency. Analysis of FL spectra and fitted data revealed a fluorescence decay time of 3.36 ns for TAPFc-TFPPy-CMP and 1.77 ns for TCNQ@TAPFc-TFPPy-CMP, indicating faster carrier separation and transport after the doping (Figure 4b). The reduction in FL suggests increased separation efficiency of photogenerated electron–hole pairs [46]. Photocurrent response tests show a notable increase in the photocurrent of TCNQ@TAPFc-TFPPy-CMP (Figure 4c), confirming improved separation efficiency of electron–hole pairs in visible light due to the TCNQ-CMP interaction. The photocurrent density significantly increased from 0.19 μA cm^−2^ to approximately 0.65 μA cm^−2^ after TCNQ complexation, representing a 3.4-fold enlargement compared to the pure TAPFc-TFPPy-CMP. This increase serves as a strong indication of TCNQ’s role in enhancing photogenerated electron transport within the composite photocatalysts and reducing the likelihood of photogenerated carrier complexation. Consequently, the photocatalytic performance of the photocatalysts is greatly improved. Electrochemical impedance spectroscopy (EIS) results also demonstrate a significant decrease in the Nyquist curve radius, revealing a decreased charge transfer resistance of TCNQ@TAPFc-TFPPy-CMP. This means that interfacial charge transfer was enhanced after doping TCNQ (Figure 4d). TCNQ, being a potent electron acceptor, plays a crucial role in enhancing the separation and transportation of photogenerated carriers in composite photocatalysts through conjugated complexation with TAPFc-TFPPy-CMP. This interaction effectively lowers the internal resistance of the photocatalysts and minimizes the likelihood of photogenerated carrier recombination. The catalytic activity of the photocatalysts is notably improved by the significant enhancement in photocatalytic activity and apparent reaction rate following TCNQ doping.

Overall, these results reveal that TCNQ doping enhances charge transfer, leading to improved separation efficiency of electron–hole pairs and overall material properties. Based on these results, the possible catalytic process for photocatalytic CO_2_ reduction using TCNQ@TAPFc-TFPPy-CMP was presented. Under the light irradiation, we believe that the photosensitizer [Ru(bpy)_3_]^2+^ was excited, and then TEOA donated one electron to the excited [Ru(bpy)_3_]^2+^ to produce [Ru(bpy)_3_]^+^. After that, [Ru(bpy)_3_]^+^ transferred an electron to the CMP, which in turn initiated the conversion process from CO_2_ to CO on the CMP surface [47,48,49,50,51].

## 3. Experimental Section

### 3.1. Materials and Chemicals

7,7,8,8-Tetracyanoquinodimethane (TCNQ), acetic acid (AcOH), *n*-butanol (BuOH), 1,2-dichlorobenzene (o-DCB), acetonitrile (MeCN), and 1,4-dioxane (DOX) were purchased from TCI chemicals. Benzaldehyde, tetrahydrofuran (THF), ethyl acetate, dichloromethane (DCM), N,N-dimethylformamide (DMF), triethanolamine, [Ru(bpy)_3_]Cl_2_-6H_2_O, and Na_2_SO_4_ were purchased from Adamas. A total of 5% Nafion solution was purchased from Alfa Aesar. 1,1′3,3′-Tetra(4-aminophenyl)ferrocene (TAPFc) and 1,3,6,8-tetrakis(4-formylphenyl)pyrene (TFPPy) were synthesized according to the reported procedures in the literature [45,46].

### 3.2. Synthesis of TAPFc-TFPPy-CMP

To a 25 mL reaction tube, TAPFc (55.0 mg, 0.1 mmol), TFPPy (61.8 mg, 0.1 mmol), o-DCB (5 mL), and 6 M AcOH (0.5 mL) were added. The mixture was heated and stirred at 120 °C for 24 h under nitrogen and then cooled to room temperature. The solid powder was filtered, washed with THF, and dried under vacuum at 120 °C for 2 h to give TAPFc-TFPPy-CMP as a red powder in 90% isolated yield.

### 3.3. CMP Doping with TCNQ

The doping procedures of the CMP with TCNQ were evaluated with varied solvents, TCNQ dosage, temperature, and duration time, all with a CMP dosage of 10 mg.

### 3.4. Photoelectrochemical Measurement

Firstly, the indium tin oxide (ITO) conductive glass required for the measurements was ultrasonically cleaned with ethanol and water for 30 min and dried. Then, 5 mg of fully ground sample (TAPFc-TFPPy-CMP or TCNQ@TAPFc-TFPPy-CMP) was dispersed in BuOH (2 mL) with 10 μL of 5% Nafion adhesive and sonicated for 30 min to form a homogenous slurry. A 20 uL drop of the resulting slurry was placed on the ITO and dried naturally to form a uniform film (area of 0.25 cm^2^) that served as a working electrode. The counter and reference electrodes were platinum plates (1 cm × 1 cm) and Ag/AgCl electrodes, respectively, while the electrolyte was Na2SO4 aqueous solution (0.25 M). The photocurrent response was measured by lighting the back of the working electrode with a 300 W Xe lamp (λ ≥ 400 nm). All photoelectrochemical measurements were recorded on an LVIUM electrochemical workstation.

### 3.5. Photocatalytic CO_2_ Reduction

Photocatalytic CO_2_ reductions were carried out using a Labsolar-6A online trace gas analysis system. In a quartz reactor, 10 mg of TAPFc-TFPPy-CMP or TCNQ@TAPFc-TFPPy-CMP and 20 mg of [Ru(bpy)_3_]Cl_2_-6H_2_O were dispersed in a 50 mL mixture of acetonitrile, H_2_O, and triethanolamine (3:1:1) and continuously stirred. After three vacuum degassing and CO_2_ filling cycles, the reaction system was then filled with CO_2_ until the pressure reached 80 kPa. The mixture was then exposed to irradiation from a 300 W Xe lamp (λ ≥ 400 nm) while maintaining a reactor temperature of 25 °C. Gas analysis was conducted every 1 h of light exposure using online monitoring via a gas chromatograph (GC9790 II). The recycling performance of the materials was assessed by filtering and washing the materials with water and THF, followed by photocatalytic CO_2_ reduction reactions under the same conditions.

## 4. Conclusions

This study successfully synthesized a conjugated microporous polymer (CMP) with a symmetric tetrasubstituted ferrocene (Fc) functional group as one organic monomer. The exceptional electron-donating properties of the Fc group facilitate charge transfer between the CMP and the *p*-type dopant TCNQ, leading to the formation of stable charge transfer complexes. Various characterizations confirmed a significant interaction between TCNQ and CMP, resulting in a reduction in the optical bandgap from 2.0 to 1.43 eV. This enhancement in the separation efficiency of electron–hole pairs notably improved the photocatalytic CO_2_ reduction performance, leading to an 80% increase in CO yield. The precise tuning of organic polymers demonstrates the potential for utilizing organic semiconductors in photocatalysis and supports further advancements in the field of organic semiconductors.

## Data Availability

The authors confirm that most of the data supporting the findings of this study are available within the article and its Appendix A. Raw data are available from the corresponding author (Y.G.) on request.

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
