# Peer review of "Doping Ferrocene-Based Conjugated Microporous Polymers with 7,7,8,8-Tetracyanoquinodimethane for Efficient Photocatalytic CO2 Reduction"

_molecules, 2024, doi:10.3390/molecules29081738_

Round 1

Reviewer 1 Report

Comments and Suggestions for Authors

The manuscript by Wang et al. on photocatalytic CO2 conversion is interesting, but needs some modifications before acceptance, as pointed below:

(1) CO2 conversion experiments are very challenging as the source of carbon must be clearly investigated. I do recommend the isotope labelling experiments to confirm the source of CO and CH4 (considering especially fast rate at the first 1-2 h of experiments).

(2) There are also some minor points to be improved, such as (i) The figure 4d should be drawn for the same values at both axes; (ii) "photocatalytic catalyst" sounds strange - "photocatalyst" is much better.  

Author Response

A point-by-point answer to the reviewers’ comments

The manuscript by Wang et al. on photocatalytic CO2 conversion is interesting, but needs some modifications before acceptance, as pointed below:

Question 1: CO2 conversion experiments are very challenging as the source of carbon must be clearly investigated. I do recommend the isotope labelling experiments to confirm the source of CO and CH4 (considering especially fast rate at the first 1-2 h of experiments).

Answer 1: We thank the referee for the valuable comments, and we do agree with the referee about the isotope labelling experiment. This question is very helpful for our research and we will carry out this in our future work. The first author of this manuscript has graduated and it is now a big difficulty for us to arrange others to continue his work. We hope the referee would understand our difficulty this time.

Question 2: There are also some minor points to be improved, such as (i) The figure 4d should be drawn for the same values at both axes; (ii) "photocatalytic catalyst" sounds strange - "photocatalyst" is much better.  

Answer 2: We read the manuscript very carefully and have corrected the errors in the revised manuscript. We thank the referee again!

Reviewer 2 Report

Comments and Suggestions for Authors

The present article titles “Doping Ferrocene-based Conjugated Microporous Polymers  with 7,7,8,8-Tetracyanoquinodimethane for Efficiently Photo-catalytic CO2 Reduction" is scientifically interesting and relevant to potential applications in the field of photo catalysis and might be beneficial for the scientific community / researchers working in this area. Although author, need some minor revision. The manuscript may be accepted for publication after resolving the following:

1.     Check line “by transferring electrons or protons….” It should be electron or holes.

2.     Author have to discuss about the methodology as how they run UV-DRS analysis, what is the texture of the material, solvent used etc.

3.     As the catalyst is organic, it would be advantageous to demonstrate its stability through post-FTIR analysis to ascertain whether the framework remains intact."

4.     The author is strongly advised to incorporate a more detailed discussion about the working mechanism of photocatalysis. This should include elucidating the fate of excitation upon exposure to light, delineating the pathways of reactive oxygen species, and elucidating the mechanisms involved in CO2 reduction processes."

5.     What the author believes to be superior for the photocatalytic process: organic or inorganic photocatalysts?

6.     If the author finds it appropriate, the following works could be cited to enhance the introduction section, thereby broadening the readership of the manuscript. Additionally, the author can utilize these works to elucidate the mechanism of photocatalysis."

https://doi.org/10.1007/s42452-019-0905-6 ; https://doi.org/10.1007/s41204-023-00341-w

Author Response

A point-by-point answer to the reviewers’ comments

The present article titles “Doping Ferrocene-based Conjugated Microporous Polymers with 7,7,8,8-Tetracyanoquinodimethane for Efficiently Photo-catalytic CO2 Reduction" is scientifically interesting and relevant to potential applications in the field of photo catalysis and might be beneficial for the scientific community/researchers working in this area. Although author, need some minor revision. The manuscript may be accepted for publication after resolving the following:

Question 1: Check line “by transferring electrons or protons….” It should be electron or holes.

Answer 1: We thank the referee for the valuable and positive comments about this work. We have corrected this in the revised manuscript.

Question 2:  Author have to discuss about the methodology as how they run UV-DRS analysis, what is the texture of the material, solvent used etc.

Answer 2: We have given a detailed discuss about the methodology and also provided the information about the texture of the material, solvent used.

Answer 3: As the catalyst is organic, it would be advantageous to demonstrate its stability through post-FTIR analysis to ascertain whether the framework remains intact."

Answer 3: We carried out this experiment before and found that there is no obvious change in FT-IR before and after the catalysis, suggesting good chemical stability. Please see the spectra below. We have described this in the revised manuscript.

Answer 4: The author is strongly advised to incorporate a more detailed discussion about the working mechanism of photocatalysis. This should include elucidating the fate of excitation upon exposure to light, delineating the pathways of reactive oxygen species, and elucidating the mechanisms involved in CO2 reduction processes."

Answer 4: We thank the referee again and we agree with the referee at this point. We have given a simple analysis about the mechanism of the photocatalysis in the revised manuscript. It is as following “Under light irradiation, the photosensitizer [Ru(bpy)3]2+ is excited, and then TEOA donates one electron to the excited [Ru(bpy)3]2+ to produce [Ru(bpy)3]+. After that, [Ru(bpy)3]+ transferred an electron to CMP, which in turn initiated the conversion process from CO2 to CO on the CMP surface.”.

Answer 5: What the author believes to be superior for the photocatalytic process: organic or inorganic photocatalysts?

Answer 5: So far, various photocatalysts have been explored. Among these photocatalysts, inorganic semiconductors have been widely employed due to their relatively high efficiency, low-cost, good robustness and easy availability. However, they have wide bandgaps, which result in a high photoexcitation energy requirement and a low utilization efficiency of sunlight. Moreover, the photogenerated carriers excited by the inorganic photocatalyst recombine easily, thus limiting the development of single inorganic photocatalysts in the field of photocatalysis. Conversely, organic photocatalysts have narrow bandgaps, which require less photoexcitation energy and have a higher utilization efficiency of sunlight. In addition, the band structure of organic photocatalysts can be tuned by tailoring the functional groups of the molecular structure. However, organic photocatalysts are structurally unstable and have a low electron transport capacity. Our study demonstrates that TCNQ can effectively modulate the band gap of organic semiconductors, thereby enhancing the catalytic properties of the material, a capability not present in inorganic semiconductors.

Answer 6: If the author finds it appropriate, the following works could be cited to enhance the introduction section, thereby broadening the readership of the manuscript. Additionally, the author can utilize these works to elucidate the mechanism of photocatalysis."

https://doi.org/10.1007/s42452-019-0905-6; https://doi.org/10.1007/s41204-023-00341-w

Answer 6: We have added the literatures in the revised manuscript, as suggested by the referee.

Round 2

Reviewer 1 Report

Comments and Suggestions for Authors

well done

Author Response

We thank the referee for the positive comments